# Virus lasers for biological detection

John E. Hales [1], Guy Matmon [2], Paul A. Dalby [1], John M. Ward [1] & Gabriel Aeppli[2,3,4]

The selective amplification of DNA in the polymerase chain reaction is used to exponentially increase the signal in molecular diagnostics for nucleic acids, but there are no analogous techniques for signal enhancement in clinical tests for proteins or cells. Instead, the signal from affinity-based measurements of these biomolecules depends linearly on the probe concentration. Substituting antibody-based probes tagged for fluorescent quantification with lasing detection probes would create a new platform for biomarker quantification based on optical rather than enzymatic amplification. Here, we construct a virus laser which bridges synthetic biology and laser physics, and demonstrate virus-lasing probes for biosensing. Our virus-lasing probes display an unprecedented > 10,000 times increase in signal from only a 50% increase in probe concentration, using fluorimeter-compatible optics, and can detect biomolecules at sub-100 fmol mL$^{-1}$ concentrations.

[1] Department of Biochemical Engineering, University College London, Bernard Katz Building, Gower Street, London WC1E 6BT, UK. [2] Paul Scherrer Institut, Villigen PSI CH-5232, Switzerland. [3] Department of Physics, ETH Zürich, Zürich CH-8093, Switzerland. [4] Institut de Physique, EPFL, Lausanne CH-1015, Switzerland. Correspondence and requests for materials should be addressed to J.E.H. (email: john.hales@ucl.ac.uk) or to G.A. (email: aepplig@ethz.ch)

Antibody-based probes tagged for fluorescent quantification can bind a wide range of biomolecular targets with high specificity, but generate a weak signal, which can be difficult to distinguish from background noise.[1] Lasing detection probes would be a superior alternative, as they would generate intense, monochromatic signals and the onset of lasing would coincide with a sharp increase in the radiant energy in a resonator-dependent emission line.[2–5] However, no new detection probes have been developed which can both harness lasing and also bind a broad range of epitopes with the specificity of antibodies, despite the construction of lasers from biological components such as DNA scaffolds and fluorescent-protein-expressing cells,[6–24] and the synthesis of plasmonic nanolasers.[25,26]

Biolasers, in which biological components form part of the lasing medium, offer greater potential for biocompatibility than plasmonic nanolasers, which are interesting and complex metallic systems.[25,26] Dye-labelled antibodies themselves are not ideal candidates because they can only typically be conjugated with up to four dyes per antibody before the degree of labelling interferes with target binding and the yield of the dyes.[27] By contrast, filamentous bacteriophage M13—a 7 nm × 900 nm rod-like virus that infects F pilus expressing strains of *E. coli*[28,29]—has been used effectively as a substitute for antibody probes in cell imaging, flow cytometry and enzyme-linked immunosorbent assays (ELISA)[30,31] and as the key component in nanosystems such as virus-based lithium-ion batteries and piezoelectric generators.[32–34] Using phage display, M13 can be routinely programmed to bind specific target biomolecules either by display of known antibody domains or binding proteins fused to either the gene 3 or 8 coat proteins, or by selection from phage-displayed combinatorial libraries to isolate those with the required target-binding specificity and affinity.[30,32,33,35–38] Phage surfaces can be functionalised by coupling amine-reactive dyes to α-amines at Ala1 and ε-amines at Lys8 on the 2700 50 amino-acid alpha-helical gene 8 coat proteins, or to solvent-exposed primary amines on the five gene 3 coat proteins.[28,29] The gene 8 coat proteins form an overlapping quasicrystalline lattice with a 5-fold rotation axis and a two-fold screw axis,[39] which brings the dyes into close proximity—much smaller than the wavelength of visible light—enabling electronic interactions and leading to resonant energy transfer between dye molecules as well as fluorescence quenching (Fig. 1a).[40–42]

Here, we create a virus laser from fluorescein-dye-labelled M13 and demonstrate its capabilities as an analytical platform for ligand-binding assays. Dye-labelled phage-clones derived from phage-display are already known to maintain their target-binding capability after selective labelling, and so these probes can replace dye-conjugated primary antibodies in biological assays.[31] We show that probes of this type can be used as virus-lasing media to generate a lasing signal in an optical configuration compatible with ordinary fluorimeters, and also to detect the binding of a target antibody molecule. The structural order and repeating chemical landscape on the surface of M13 provide a versatile and amenable model system for modulating the spectrum and threshold dynamics of the laser using the principles of synthetic biology. At the transition to lasing, the photon flux from the probes increases by five orders of magnitude, the spectral linewidth narrows to below 5.0 nm, and the sensitivity of the output intensity to small changes in the probe concentration or environment is heightened. We show that a 50% increase in the concentration of the probes from 100 pmol mL$^{-1}$ results in a >10,000 times increase in signal, which is the greatest responsivity to probe concentration shown in any biochemical assay to the best of our knowledge. In a proof-of-concept study, we optically engineer a mix-and-measure ligand-binding assay sensitive to 90 fmol mL$^{-1}$ monoclonal antibody, suggesting that clinically relevant concentrations of biomolecules can be detected without immobilization of the ligand or probe on surfaces and without invasive wash steps.

## Results

**Virus lasers as an analytical platform**. We examined the properties of virus lasers in which the gain medium is a solution of fluorescein-dye-labelled M13 in two contrasting resonator geometries. Resonant cavity R1 was constructed to explore virus-lasing for metrology (Supplementary Fig. 1), and resonant cavity R2 was engineered to strengthen the coupling of the virus-lasing to the near-field and electronic interactions of the dyes (Fig. 1b, Supplementary Fig. 1). In both instruments, the excitation and detection optics were similar to those in a fluorimeter, except for the greater than eight-decade dynamic range of the spectrometer and the intensity of the 3–5 ns excitation pulses at 493 nm (Fig. 1b; details in Methods). The gain medium was circulated between a reservoir and a flow cuvette and the pump light source operated in single-shot mode to allow the dye to be refreshed between pulses (Fig. 1b).

R1 consisted of a flat mirror separated by 300 mm from a spherical mirror with a radius of curvature of 400 mm, equivalent to 2 round-trips per FWHM (full width at half maximum) of the temporal profile of the pump laser pulse. The diameter of the beam was constrained only by the 2 mm × 2 mm windows of the cuvette. In contrast to microresonators, these dimensions are much greater than microbiological length scales ensuring that there was no mechanical interference from the resonator.

In Fig. 2a, we display a series of threshold curves demonstrating laser action for different probe concentrations as well as for fluorescein in solution with no virus. For the highest probe concentration (purple), 413 pmol mL$^{-1}$ M13 conjugated with 386 dyes per M13 (V1)— equivalent to 0.7 dyes on average per ring of proteins—doubling the pump energy from below to above the threshold point for lasing, increased the intensity at the spectral peak by a factor of 16,000 (Fig. 2a).

We have measured the threshold region of the virus laser with unprecedented precision across 7 decades of intensity because it is this non-linearity that makes lasing such a unique analytical prospect. Data for other biolasers have typically been restricted to 1–2 decades of dynamic range[6–9,11–13,15–24] (Supplementary Fig. 3) and quite often, only the above-threshold region was measured,[6,11,13–17,19–24] meaning that the position of the threshold point could only be found from linear extrapolation, reducing the potential utility of these biolasers as analytical instruments. In Fig. 2a, the initial deviations in intensity from sub-threshold behaviour are due to the amplification of seed fluorescence before the gain can overcome the resonator losses, and the inflection points are the transitions to lasing.

The threshold for 146 nmol mL$^{-1}$ fluorescein (F1, dark orange) was 2.2 times lower than for 351 pmol mL$^{-1}$ V1 (dark blue), which had a similar volume-averaged optical density equivalent to 135 nmol mL$^{-1}$ fluorescein. This implies that there was enhanced quenching of the dyes attached to M13 due to dye–dye or dye–protein interactions, since the lasing threshold $\varphi_{th}$ varies according to

$$\varphi_{th} = \frac{K_2 K_L}{K_F(D K_{RAD} - K_L)} \quad (1)$$

This equation is derived in the Theoretical models section of the Methods. The microenvironments provided by the virus and solvent for the dyes will affect the decay rate $K_2$ from the upper state of the dye, but optical absorption data show no influence on

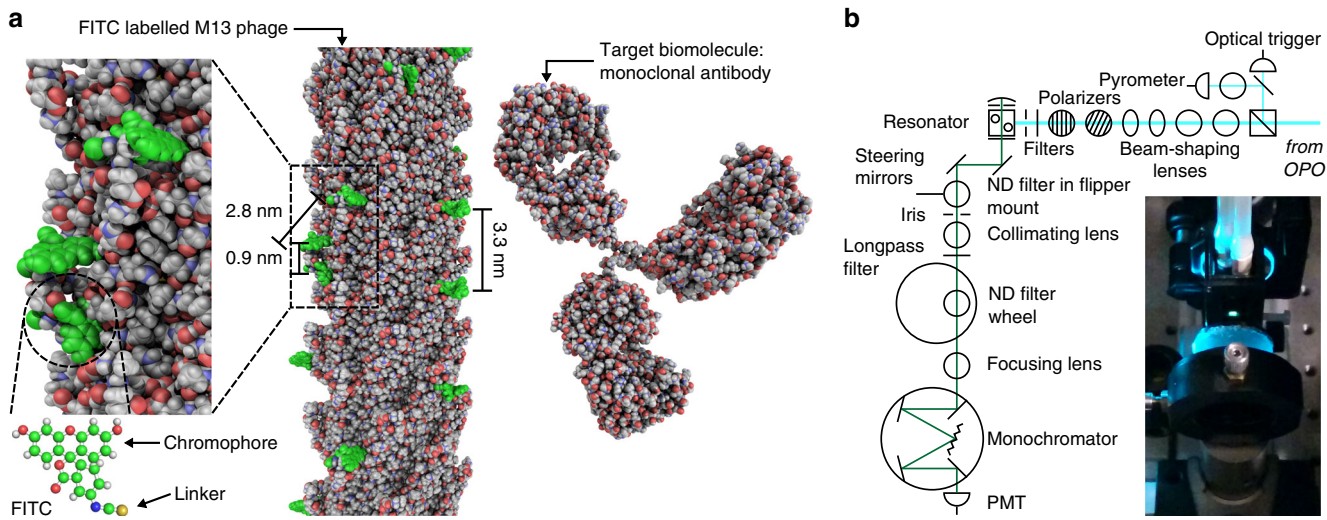

**Fig. 1** Design of the virus laser. **a** Model of the atomic structure of a lasing detection probe composed of M13 bacteriophage (PDB: 2MJZ) covalently modified with fluorescein isothiocyanate isomer 1 (FITC) dyes (PubChem CID: 18730, green).[29,57,58] There are ~540 rings of gene 8 coat proteins per M13, but only 13 are shown here. The target biomolecule is a IgG2a monoclonal antibody (mAb) that can be bound by the gene 3 coat proteins and is represented in the figure by the structure of an intact IgG2a mAb (PDB: 1IGT) which should have a homologous structure.[59] Rectangular zoomed area, image of the surface of M13 showing the close proximity of the attached dyes. Circular zoomed area, chemical structure of fluorescein isothiocyanate. Fluorescein has shorter moieties bonded to the aromatic ring system than other xanthene dyes, such as rhodamine 6G, reducing the likelihood of contact quenching between neighbouring dyes. **b** Optical configuration for experiments conducted using resonant cavity R2 (Supplementary Fig. 1). The excitation source was an optical parametric oscillator (OPO), the detector was a photomultiplier tube (PMT) and the emission was attenuated by neutral density (ND) filters. Inset, photograph of the emission from the gain medium of the virus laser (green light) as viewed from along the axis of the resonator

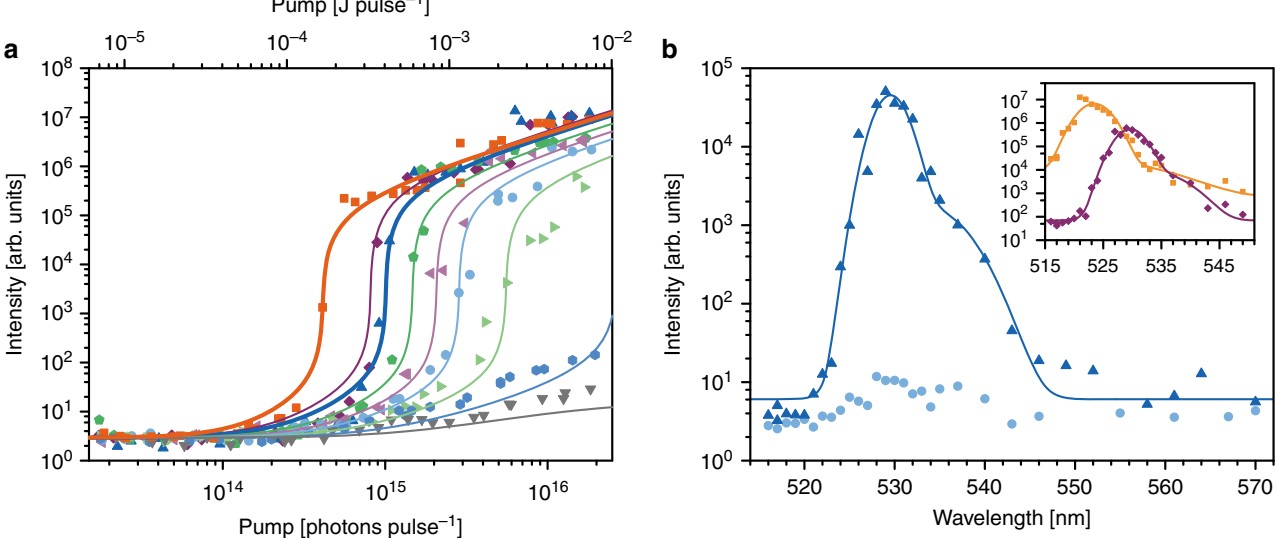

**Fig. 2** Virus lasers. **a** Threshold behaviour in R1 for 146 nmol mL$^{-1}$ fluorescein (F1, dark orange) and for virus-lasing probes (V1) diluted from 413 pmol mL$^{-1}$ (purple) to 351 pmol mL$^{-1}$ (dark blue, equivalent to 135 nmol mL$^{-1}$ fluorescein), 270 pmol mL$^{-1}$ (green), 219 pmol mL$^{-1}$ (light purple), 185 pmol mL$^{-1}$ (cyan), 140 pmol mL$^{-1}$ (light green), 103 pmol mL$^{-1}$ (blue), and 72 pmol mL$^{-1}$ (grey). The fitted curves represent a global fit of all of the sets of threshold data to a theoretical model (Supplementary Fig. 5; Methods). Errors in the concentration described in Methods. **b** Emission spectra of V1 at 413 pmol mL$^{-1}$ as pumping increased from $5.5 \times 10^{14}$ photons pulse$^{-1} \pm 10.1\%$ (cyan) to $9.1 \times 10^{14}$ photons pulse$^{-1} \pm 6.6\%$ (dark blue). Inset, bathochromic shift in the spectrum of V1 (purple, pumping of $1.5 \times 10^{15}$ photons pulse$^{-1} \pm 10.9\%$) relative to 72 nmol mL$^{-1}$ fluorescein (orange, $1.7 \times 10^{16}$ photons pulse$^{-1} \pm 11.4\%$). Curves are fits to the sum of two Gaussian peak functions. The intensity was not calibrated but was proportional to the number of photons entering the spectrometer (see Methods). Source data are provided as a Source Data file

the strength $K_F$ of the coupling of the pump to the dyes (Supplementary Fig. 2). If the resonator, defined by the loss rate $K_L$, the rate $K_{RAD}$ of spontaneous emission per dye into the resonator mode and the number $D$ of dyes in the mode volume are unchanged and $D$ is large, the ratio of ultimate lasing output energy $n$ to pump power $\varphi$ will be the same for V1 and F1

because in the high power limit

$$\frac{n}{\varphi} \to \frac{\beta K_F}{K_{RAD}} \left( \frac{D K_{RAD}}{K_L} - 1 \right) \to \frac{\beta K_F D}{K_L} \quad (2)$$

where $\beta$ is the optical attenuation due to the output coupler and

the spectral bandwidth of the spectrometer (equations derived in Methods). The data in Fig. 2a are consistent with equation (2).

The curves in Fig. 2a were derived from a model of the threshold dynamics that describes the energy build-up in the dominant mode in terms of the pump energy and the dye concentration while accounting for the seed fluorescence (see Methods). We fit the data globally with shared parameters for experimental variables retained between measurements, such as the number of resonator modes, and fixed values for variables which could be calculated in advance, including the rate of loss from the resonator. From this model and by noting that from equation (1) the threshold point is proportional to $\left(\frac{D}{D_{min}} - 1\right)^{-1}$ where $D_{min} = \frac{K_L}{K_{RAD}}$, we estimate that the minimum number of probes required for lasing is 93.4 pmol mL$^{-1}$ ± 0.7 pmol mL$^{-1}$, which is approximately three orders of magnitude lower than the lowest reported DNA Förster resonance energy transfer (FRET) laser probe concentration.[24] This value is at the upper limit of the useful range for probes in a clinical context, but it could be reduced straightforwardly by decreasing $K_L$ and/or increasing $K_{RAD}$ via steps such as minimising Fresnel reflections at the cuvette side-walls and shortening the resonant cavity.

Spectral linewidth narrowing has not been shown for a solution-state biolaser previously and even here the spectral measurement below threshold has been hampered by low light levels. Even so, Fig. 2b demonstrates that the sharp rise in intensity at threshold coincided with a change in the spectral line-shape and a narrowing of the spectral linewidth to 4.0 nm ± 0.3 nm (FWHM) above threshold at a pump energy of $9.1 \times 10^{14}$ photons pulse$^{-1}$ ± 6.6%.

Unlike for a DNA FRET laser or a green fluorescent protein cell laser, the biological structural order of M13 impacts the spectroscopic characteristics of a homogeneous ensemble of chromophores in the virus laser. Moving the dyes closer together first by increasing the density in solution and then via immobilization on the viruses red-shifted the spectral peak from 523.4 nm ± 0.2 nm for 72 nmol mL$^{-1}$ fluorescein (mean separation 28.5 nm) to 527.2 nm ± 0.4 nm for 301 nmol mL$^{-1}$ fluorescein (17.7 nm) to 529.6 nm ± 0.2 nm for V1 (Fig. 2b inset, Supplementary Fig. 4). Thus, lasing viruses are analogous to genetically-programmable quantum dots whose spectral peak emission can be tuned not by varying their size but by varying the number of dyes attached per virus.

Figure 3a and b show that the concentration of dye-labelled M13 probes can be determined from the position of the threshold point. We derived a simple function that can model threshold data without requiring precise knowledge of the experimental variables, and in Fig. 3a we implement this function (equation (26) in Methods) to calculate the threshold points as a function of the probe concentration (dark blue), and in matching units for freely-dissolved fluorescein (dark orange) (see Supplementary Fig. 6 for individual fits). The data points are in excellent agreement with the cyan curve, which is the expected threshold point position based on the global model of the threshold dynamics (see Fig. 2a), proving that each set of threshold data contained enough information to allow the probe concentration to be calculated precisely.

Typically, threshold data are plotted as the intensity against pump energy at constant dye concentration, as is the case in Fig. 2a, but in Fig. 3b we show that the same data can be plotted as the intensity against probe concentration at constant pump energy. Mirroring a conventional threshold curve, the output intensity was extremely sensitive to small changes in the probe concentration when the pump energy was close to the associated threshold point. For instance, from the model globally fit to the

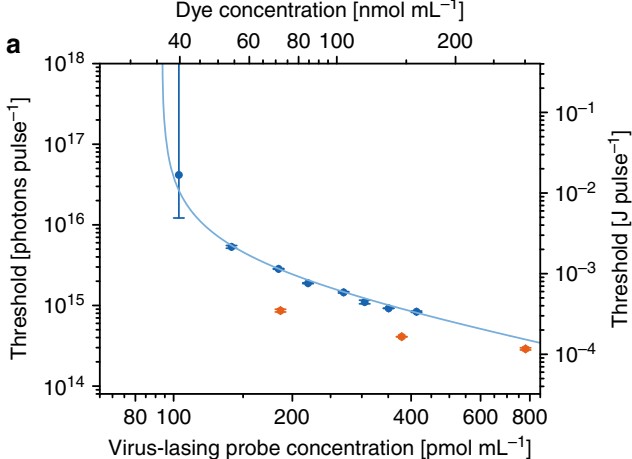

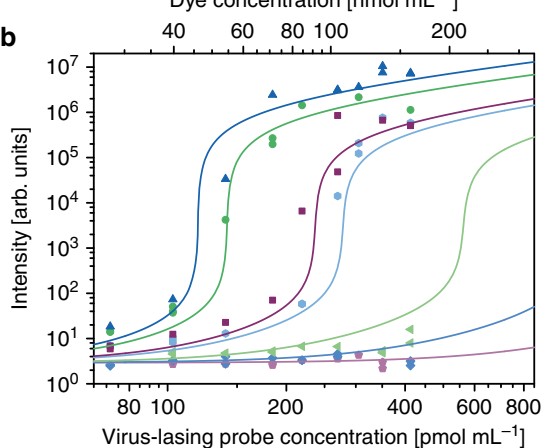

**Fig. 3** Quantification of virus-lasing probes. **a** Response of the threshold point to variations in probe concentration. Scatter points represent the threshold point calculated by fitting individual sets of threshold data of either V1 (dark blue) or fluorescein (dark orange) using equation (26). The line is the expected dependence of the threshold point on the probe concentration calculated from the global fit in Fig. 2a. The error bars are standard errors and the error bar that extends beyond the upper limit of the y-axis extends to infinity. **b** The intensity at different probe concentrations has been plotted at pump energies of $1.0 \times 10^{16}$ photons pulse$^{-1}$ ± 5.7% (dark blue), $5.4 \times 10^{15}$ photons pulse$^{-1}$ ± 7.6% (green), $1.8 \times 10^{15}$ photons pulse$^{-1}$ ± 6.2% (purple), $1.4 \times 10^{15}$ photons pulse$^{-1}$ ± 5.3% (cyan), $5.5 \times 10^{14}$ photons pulse$^{-1}$ ± 4.8% (light green), $2.5 \times 10^{14}$ photons pulse$^{-1}$ ± 6.3% (blue), $9.6 \times 10^{13}$ photons pulse$^{-1}$ ± 5.8 % (light purple). The lines are simulations from the same global fit for the intensity against probe concentration at the same pump energies. Source data are provided as a Source Data file

data, if the pump energy were fixed at $1.001 \times 10^{16}$ photons pulse$^{-1}$, which was the threshold point for 119.4 pmol mL$^{-1}$ V1, a 50% increase in the probe concentration from 100.0 pmol mL$^{-1}$ to 150.0 pmol mL$^{-1}$ would result in a 17,000 times increase in the intensity. Away from the threshold point, a 50% change would simply result in a ~50% change in the intensity, as is typical for a ratiometric assay, which might be difficult to see above the background. This is the greatest responsivity in output intensity to small changes in probe concentration that has been recorded in any biological assay, as far as we know. The analytical concept has immediate applications in tipping point measurements, where the sample would be pumped at the threshold intensity for the critical

concentration of lasing probes to yield an unambiguous, digital readout: if there is lasing, then the critical concentration is exceeded, otherwise there is no lasing. Alternatively, the pump energy could be scanned to find the threshold point, and therefore the probe concentration, via cross-referencing with a calibration plot such as Fig. 3a.

We confirmed that the probe could still infect F pilus strains of E. coli by performing a plaque titer assay which is a technique used to estimate the concentration of infectious phage (details in Methods). In this assay, the target molecules are the bacterial F pilus and then the membrane protein TolA on the surface of E. coli which the g3p protein must bind for the phage to infect the cell.[43] The titer assay yielded $10^{13}$ plaque-forming units mL$^{-1}$ (Supplementary Table 1), substantiating that the dye-labelled M13 remained infectious and therefore suggests that g3p was not inhibited from binding its receptor by the covalent modification of the coat proteins with fluorescein dye. Furthermore, dye-labelled mutant M13 have been shown to retain their binding selectively for granulocytes and monocytes in a g3p-based phage display system in both flow cytometry and immunofluorescence studies.[31]

As for other optical assays, control measurements prior to the addition of the lasing detection probes can fix the contribution of noise sources and any other chromophores in the sample. Beyond this, the potential for multiplexing in virus lasers is intrinsically greater than for fluorescence-based techniques such as flow cytometry because the emission spectral linewidths are narrower, enabling the implementation of companion probes to monitor changes in environmental factors for even greater measurement precision. To do this, we propose decomposing the overlapping gain spectra of the different probes by sweeping the dispersive element in a tunable virus laser to record the intensity at different emission wavelengths. Future work will aim to quantify the multiplexing capability of virus lasers, determine the impact of cellular milieu and blood serum on lasing, and harness synthetic biology to engineer environmentally insensitive probes.

**Virus-laser mix-and-measure ligand-binding assay.** Most assay formats gain sensitivity by trading a degree of disruption to the biological system, for instance, by immobilising the ligands or probes and introducing wash steps, or by labelling the ligands. Given the unprecedented signal-generation capabilities of virus-lasing probes, we have tested the feasibility of a mix-and-measure ligand-binding assay which does not necessitate these trade-offs. Analogous to a plasmonic approach where ligand-binding to antibodies immobilized on a metal surface modulates the optical response,[44] the aim was to measure shifts in the threshold point and variations in the output intensity in response to solution-phase binding of the lasing probes. For such an assay, the virus laser needs to be sensitive to the binding of the probes to a target ligand. Accordingly, we increased the impact of near-field and electronic interactions occurring at sub-1 ns timescales on the surface of the phage by reducing the resonator (designated R2) length to 26 mm, corresponding to a round-trip time of 195 ps taking the refractive index of the gain medium into account (Fig. 1b). We increased the effect of binding on a single probe by increasing the density of dye molecules per probe, and reduced the concentration of probes to less than $D_{min}$ for resonator R1. This improvement in the sensitivity resulted from the reduction in the resonator length, which increased $K_{RAD}$; further performance enhancements would result from extending this approach. The number of spatial modes was constrained by introducing a 300 µm pinhole so that the Fresnel number of the resonator was $F = 6.7$.

For both 23 pmol mL$^{-1}$ M13 conjugated with 564 dyes per M13 (V2), equivalent to 1.0 dye on average per ring of coat proteins, and 13 nmol mL$^{-1}$ unbound fluorescein dye molecules in buffer (F2), threshold and spectral measurements were acquired before and after (Fig. 4a, b, Supplementary Fig. 7 and

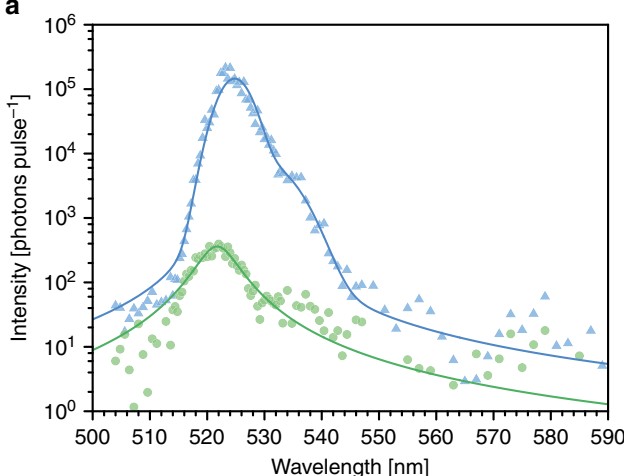

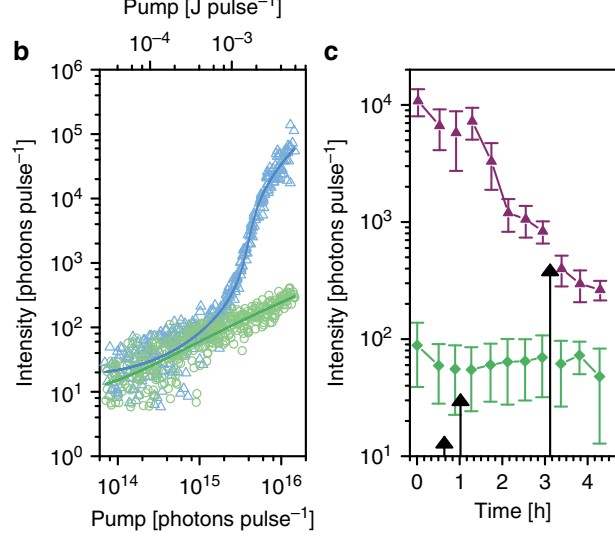

**Fig. 4** Response of the virus laser to ligand-binding. **a** Spectra of V2 before (cyan) the addition of cp-mAb at $1.3 \times 10^{16}$ photons pulse$^{-1}$ and after (light green) the addition of the antibody at $1.4 \times 10^{16}$ photons pulse$^{-1}$. Each scatter point is the mean of four measurements at each wavelength and the curves are fits to the sum of two Voigt peak functions (dark blue) and a Lorentzian peak function (green). **b** Threshold behaviour of V2 before and after the addition of cp-mAb, same colours as (**a**). Lines represent individual fits of the data to equation (26). **c** Time-dependence of the intensity at 523.5 nm at pump energies initially above threshold at $1.3 \times 10^{16}$ photons pulse$^{-1}$ ± 4.7% (purple) and below threshold at $1.6 \times 10^{15}$ photons pulse$^{-1}$ ± 4.4% (green). The concentration of cp-mAb, which is a ligand bound by M13, in the gain medium after mixing increases in steps from 0 fmol mL$^{-1}$ to 9 fmol mL$^{-1}$, 29 fmol mL$^{-1}$ and then 90 fmol mL$^{-1}$. The arrows indicate when cp-mAb was added to the reservoir, and the lengths of the arrows are proportional to the amount added to the reservoir at each step. The squares and error bars represent the mean and standard deviation, respectively, of 7 measurements taken at each point in the time series. The variation in intensity at each time point is caused by small fluctuations in the pump energy and random variations in the response of the intensity to pumping. See Supplementary Figs. 7, 8 and 9 for all of the collected data. Source data are provided as a Source Data file

Supplementary Fig. 8) the addition of a monoclonal antibody (cp-mAb, illustrated in Fig. 1a) to the reservoir feeding the gain medium, which is a ligand bound specifically by the five gene 3 coat proteins at one end of the phage. Between these measurement sets, for V2 only, we collected intensity data at three different levels of pumping 15 min after adding either a buffer control or cp-mAb into the reservoir containing the detection probes (Fig. 4c, Supplementary Fig. 9).

For V2, lasing could no longer be sustained with pump pulses of $1.4 \times 10^{16}$ photons pulse$^{-1}$ after the addition of 90 fmol mL$^{-1}$ cp-mAb to the gain medium, resulting in a 690-fold decrease in the intensity at 524.8 nm (Fig. 4a). The cp-mAb was added to the reservoir feeding the gain medium in three steps, so the intensity data reveal the response of the virus laser to step increases in cp-mAb. The addition of 9 fmol mL$^{-1}$ cp-mAb to the gain medium and then another 20 fmol mL$^{-1}$ cp-mAb triggered an increase in the threshold point of V2 as the antibody mixed and bound to the surface of the lasing detection probes (Fig. 4c). The threshold point reached a new steady-state position after ~70 min with no subsequent addition of cp-mAb, but this was disturbed upon addition of a further 61 fmol mL$^{-1}$ cp-mAb, which coincided with the cessation of lasing (Fig. 4c).

The initial 5.5-fold step-change at a pump level of $1.3 \times 10^{16}$ photons pulse$^{-1}$ was in response to 29 fmol mL$^{-1}$ cp-mAb, which equates to 780 lasing detection probes per 1 target biomolecule, and when pumping was below-threshold the addition of cp-mAb had no effect on the intensity, indicating that lasing was crucial to the responsivity (Fig. 4c). The antibodies bound to the phage would red-shift the absorption spectra of certain dyes due to solvent exclusion and contact with a dielectric body with polar side-chains:[45,46] anti-fluorescein fab fragments bound to fluorescein shift the absorption peak from 491 nm to the ranges 505 nm to 507 nm and/or 518 nm to 520 nm with fluorescence quenching maxima values of 95% to 97%.[47] Theoretical studies of 5 µM mL$^{-1}$ donor dye in bulk solution doped with 0.1 µM mL$^{-1}$ acceptor dye suggest that these red-shifted dyes would sink energy by cavity-enhanced energy transfer and FRET, increasing the threshold point.[48] This effect would be enhanced by homo-FRET transfer between dyes on the same phage,[49] and absorption of the emission by ground-state dyes on the antibody-bound phage. Our experiment provides the first empirical evidence that a biolaser can detect a medically-relevant class of biomolecule at clinically relevant concentrations below the probe concentration.

Since the fraction of ligands bound by probes initially increases at a rate determined by the probe concentration and the on rate, which is typically ~$10^5$ M$^{-1}$ s$^{-1}$ for an antibody,[50] mix-and-measure ligand-binding assays with a large excess of probes could rapidly detect sub-1 fmol mL$^{-1}$ ligand concentrations, which is often not practical for nanosensors or surface-based methods due to long accumulation times.[51] The characteristic timescale for shifts in the threshold point at a probe concentration of 23 pmol mL$^{-1}$ would be 7.2 min if the on rate were ~$10^5$ M$^{-1}$ s$^{-1}$ [50] so the effect of target-binding could be readily distinguished from effects that would cause an immediate shift in the threshold point, including changes in the optical density of the gain medium or the quality factor of the resonant cavity.

ELISA, which involves binding the ligands to surface-immobilized antibodies before adding a large excess of a second antibody which binds a different epitope of the surface-bound ligands, can achieve detection limits between 1 pmol mL$^{-1}$ and 0.1 amol mL$^{-1}$ depending on the antibody-antigen affinity,[52] but unbound probes must be washed away making this technique laborious, disruptive to the biological system and unsuited to continuous monitoring applications. Here, sources of systematic error could be mitigated using strategies analogous to those accepted as best-practice for surface-plasmon-resonance ligand-binding assays. For example, a differential measurement could be set up by adding the biological sample to matched solutions containing different phage-clones with the same density of fluorescein per probe but with g3p coat proteins programmed to either bind or not bind the target ligand. The accuracy would depend on the precision with which the lasing threshold could be measured, which would be very high given that this simply involves measuring the photon flux of the pump, and the similarity of the initial lasing responses of the binding and non-binding probe solutions. In the control experiment, F2 was only marginally responsive to the addition of 9.1 pmol mL$^{-1}$ cp-mAb, showing that fluorescein does not bind cp-mAb and that the lasing performance is not inherently affected by the target biomolecule or by any other molecules or salts in the solution (Supplementary Fig. 7, Supplementary Fig. 8). Other explanations for the observed change in the threshold dynamics, including photobleaching, dilution of the probes due to the filter and the addition of buffer, and air bubbles have been ruled out for reasons explained in the Supplementary Discussion.

In two further experiments, photobleaching caused a steady increase in the threshold point of the virus laser as time elapsed without any step-changes, and in further experiment 1, there was no response to the addition of a non-binding control antibody (mouse IgG2a isotype control) (Fig. 5). In both experiments, the subsequent addition of the cp-mAb ligand resulted in a step-change in the threshold point confirming the specificity of the ligand-binding assay. The effect is more pronounced and immediate in further experiment 2 than in further experiment 1, consistent with more cp-mAb being adding in further experiment 2. See the Methods for more experimental details and the Supplementary Discussion for further analysis.

If the gain were insufficient to overcome the losses due to absorption or scattering by non-target molecules, or if the losses were too variable to accurately measure the threshold point, then we could dilute the sample in a buffer solution as is common practice in spectrophotometry. For instance, R2 could be used as a process analytical technology for detecting antibodies in samples extracted directly from an industrial bioreactor if the sample were diluted 10,000-fold in the buffer reservoir feeding the gain medium. If the cell culture had an optical density of 100 at the emission wavelength, then the round-trip loss after dilution would be 2% from the Beer–Lambert law, which would not prohibit lasing. We have shown that R2 is sensitive to 29 fmol mL$^{-1}$ cp-mAb so we estimate that ~300 pmol mL$^{-1}$ IgG in an industrial bioreactor could be detected using this approach. For comparison, the mean of the central 95% range of the age-corrected human serum IgG concentration distribution is 70 nmol mL$^{-1}$.[53]

## Discussion

Virus biolasers can deliver a five decade increase in signal, 4 nm emission linewidths and large, non-linear responses in output intensity to small changes in probe concentration. The optical technologies required for lasing detection, including compact pump lasers and photodetectors, could be readily integrated into existing fluorimeter platforms, making them into powerful tools for digital biomedicine. Our proof-of-concept experiments suggest that biolasers could measure clinically relevant concentrations of biomolecules with a sensitivity approaching that of ELISA, yet without the need for probe immobilization or multiple wash steps. This paves the way for research, eventually employing whole blood and urine with clinical applications in mind, to genetically engineer a series of virus-lasing probes which selectively bind target biomolecules directly in solution.

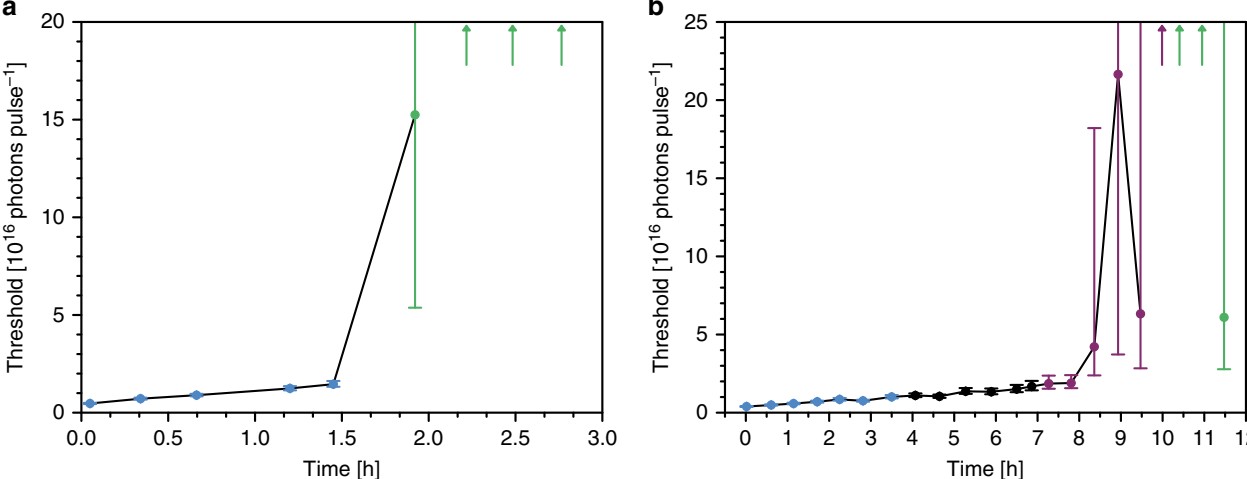

**Fig. 5** Step-changes in the threshold point in response to ligand-binding. In two further experiments in R2, the threshold behaviour of dye-labelled M13 was monitored as a function of time and the addition of cp-mAb or a non-binding antibody. For each experiment, the data were fit globally using equation (26) with parameters $\chi_0$ and $\chi_1$ shared between each set of threshold measurements. For further experiment 2, 20 pmol mL$^{-1}$ V2 contained no antibody (blue), before 4.5 pmol mL$^{-1}$ cp-mAb was added (green). For further experiment 1, 23 pmol mL$^{-1}$ V2 initially contained no antibody (blue), 91 fmol mL$^{-1}$ mouse IgG2a isotype control was added (black) and then cp-mAb was added in two steps so that the concentration of cp-mAb was initially 91 fmol mL$^{-1}$ (purple) and then increased to 1.9 pmol mL$^{-1}$ (green). See Methods for more details. The threshold points derived from the fitted models have been plotted against time for, **a** further experiment 2 and for, **b** further experiment 1. Measurements have been represented by an arrow if the fitting implied that the sample could no longer sustain lasing. The error bars represent standard errors and the error bars that extend beyond the upper limit of the y-axis extend to infinity. See Supplementary Fig. 10 for an expanded version of this figure with the intensity against pump energy data for different points in the time series. Source data are provided as a Source Data file

## Methods

**Sample preparation.** The M13 stock was prepared from a single plaque to ensure genetic homogeneity. M13 was amplified in *E. coli* Top10F' cultures and purified by several rounds of centrifugation with or without 5% polyethylene glycol (PEG-8000) and 0.2 M sodium chloride to either selectively precipitate the phage or to remove cells and cellular debris. For the first round of precipitation for the M13 prepared for experiments in R1, the concentration of precipitating agent was reduced to 3.3% PEG-8000 and 130 mM NaCl. The M13 solutions were filter-sterilised with a 0.22 μm filter. For the experiments performed in R1, 24.9 mg fluorescein isothiocyanate isomer 1 (FITC) powder was added to 50 mL 2.0 mg mL$^{-1}$ M13, 100 mM sodium borate, pH 9.1, and incubated for 2 hr at 37 °C prior to quenching with 20 mL 1 M tris-HCl, pH 7.5. For the experiments performed in R2, 15.5 mg FITC powder was added to 29.8 mL 0.7 mg mL$^{-1}$ M13, 100 mM sodium borate, pH 9.2, and incubated for 4 h 15 min at 37 °C prior to quenching with 90 mL 133 mM tris-HCl. In both procedures, the dye was removed by rounds of centrifugation with and without PEG-8000/NaCl. The precipitated M13 was resuspended in 100 mM sodium borate, pH 9.0. For the dye-labelled M13 used in R2, the solution was also driven through a 0.22 μm filter.

The M13-binding antibody (Life Technologies, M13 phage coat protein monoclonal antibody from clone E1) which recognises the five g3p coat proteins at one end of M13 and the non-binding antibody (Life Technologies, Mouse IgG2a isotype control) were aliquoted and stored at −20 °C at a concentration of 1 mg mL$^{-1}$. Before use, the antibody stock would be thawed prior to dilution to the appropriate concentration. The antibody was added to the reservoir of the gain medium by pipetting the antibody onto the side-wall of the reservoir and washing the side-wall with the sample ejected from the silicone tubing connected to the flow cuvette. The antibody was allowed to mix with the sample solution as it circulated between the reservoir and the flow cuvette prior to measurement for 15 min for V2 and for 20 min for F2 (Fig. 4, Supplementary Figs. 7–9). The buffer control was added in the same way. The molecular mass of the M13-binding antibody was assumed to be approximately 160 kDa.

**M13 titer assay.** Dye-labelled M13 was serially diluted and 10 μL drops of each dilution were spotted onto bacterial lawns, which were prepared by pouring 4 mL of a mix of 0.7% Nutrient Broth No. 2 agar and *E. coli* Top10F' culture onto thin base 1.5% Nutrient Broth No. 2 agar plates. The mix contained 20 mL *E. coli* Top10F' culture grown overnight in Nutrient Broth No. 2 per 200 mL 0.7% Nutrient Broth No. 2 agar. After spotting the dye-labelled M13 dilutions, plates were incubated overnight at 37 °C. The phage titer was calculated from the number of plaques, the dilution factor and the sample volume.

**Free-space optical assembly.** The resonant cavities were pumped with pulses of 493 nm light with a FWHM (Full Width at Half Maximum) of 3 ns to 5 ns from an

optical parametric oscillator (EKSPLA, NT 342B-20-AW) in single-shot mode via beam-shaping optics consisting of spherical (Comar Optics, 200 PB 25 and 100 NB 25) and cylindrical (Comar Optics, 100 YB 40 and 25 UB 25) BK7 crown-glass lenses with anti-reflection coatings, which transformed the 4.5 mm diameter circular beam cross-section to a 2 mm × 9 mm ellipse (Supplementary Fig. 1). The pulse intensity was controlled with two Glan-Laser calcite polarizers (Thorlabs, GL10). The first was mounted in a motorised rotation stage (Thorlabs, PRM1/MZ8) connected to a T-cube DC servo controller (Thorlabs, TDC001) and the second was fixed with its polarization axis parallel to the polarization of the excitation pulses. The pump energy was monitored by partially reflecting the light with either, in R1, a pellicle beamsplitter (Thorlabs, BP208), or, in R2, a glass window, towards a pyroelectric probe (Molectron, J25) connected to a Joulemeter (Molectron, EM500). In R2, a plano-concave, $f = -50$ mm lens (Newport, KPC040) was placed before the pyroelectric probe to match the widths of the beam and the sensor. In R1, the detector electronics was triggered by the electronic trigger from the pump source; in R2 it was triggered by an optical trigger, consisting of a photodiode (Thorlabs, SM1PD1A) connected to a high-speed circuit preceded by an OD 2 neutral density filter (Thorlabs, NDUV20B) with minor damage to the optical coating, positioned behind a protected aluminum mirror (Thorlabs, PF10-03-G01) in the path between the beamsplitting window and the pyroelectric probe. Additionally, for R2, the pump was transmitted through a rectangular aperture that matched the dimensions of the cuvette window, and a dielectric shortpass filter with a cut-off wavelength of 510 nm (Comar Optics, 510 IK 50).

The pump impinged on a 2 mm × 8 mm window to a flow cuvette (Starna, 583.2.2FQ-10/Z15) with a 2 mm × 2 mm × 10 mm internal chamber, so the maximum path length of the pump light through the cuvette was 2 mm since the long axis of the internal chamber was orthogonal to the direction of the pump beam and parallel to the axis of the resonant cavity. The samples or cleaning fluids were flowed into the cuvette via silicone tubing from reservoirs using a peristaltic pump (RS Components), and samples would then flow back into the reservoir. The flow rates were 2.5 mL min$^{-1}$ for R1 and 3.0 mL min$^{-1}$ for R2, and the time between pulses was long enough for the dye in the cuvette to be refreshed. For R2, there was a 5 μm in-line filter (Whatman, Polydisc HD 5.0 μm) between the reservoir and the cuvette. In further experiment 2 in R2, the in-line filter was removed and the flow rate was increased to 4 mL min$^{-1}$. The cuvette had 2 mm × 2 mm Spectrosil quartz windows on opposing walls at 90° to the entrance window, which were aligned with the axis of the resonant cavity. The Fresnel reflectance at the glass interface would have been,

$$r = \left( \frac{n_1 - n_2}{n_1 + n_2} \right)^2 \qquad (3)$$

so that $r = 0.0350$ at the glass-air interface for $n(\text{fused} - \text{silica}) = 1.46$ and

$n(\text{air}) = 1.00$, and $r = 0.00217$ at the glass-water interface for $n(\text{water}) = 1.33$. Consequently, the transmission through the cuvette was nominally 0.860 per round-trip of the resonant cavity.

R1 consisted of a flat dielectric output coupler (Comar Optics, 25 MX 02) and a dielectric spherical mirror with a radius of curvature of 400 mm (Thorlabs, CM508-200-E02), separated by 300 mm with the cuvette adjacent to the output coupler. In R2, the spherical mirror was replaced with a compound optic consisting of a protected silver spherical mirror with a radius of curvature of 50 mm (Thorlabs, CM254-025-P01) bonded to a 300 µm pinhole (Thorlabs, P300S) and the separation of the mirrors was reduced to 26 mm. The reflectivity of the flat mirror was 99% and the reflectivities of the dielectric and metallic spherical mirrors were 99.6% at 530 nm unpolarized and 98.8% at 520 nm unpolarized, respectively, based on product data from the manufacturers. The fraction of the oscillation retained was then nominally 0.848 per round-trip for R1, and 0.841 per round-trip for R2. The round-trip time for light oscillating between the mirrors of the resonant cavities was 2.0 ns for R1 and 195 ps for R2, for a cuvette filled with water. The diameters of the mirrors were much greater than the diameters of the cuvette windows and the pinhole, so the Fresnel number

$$F = \frac{a^2}{L\,\lambda} \qquad (4)$$

where $a$ is the radius of the aperture and $L$ is the length of the resonant cavity, was approximately $\frac{(1\,\text{mm})^2}{300\,\text{mm} \times 530\,\text{nm}} = 6.3$ for R1 and $\frac{(0.3\,\text{mm})^2}{26\,\text{mm} \times 520\,\text{nm}} = 6.7$ for R2. In both cases, $F > 1$ so large diffraction losses are not anticipated.

The effective volume of the gain medium is the volume that is pumped which can contribute to the laser emission. For R1, this is the entire volume of the cuvette chamber that is subjected to pumping, $V_{\text{eff}} = 8\,\text{mm} \times 2\,\text{mm} \times 2\,\text{mm} = 32\,\text{µL}$. For R2, this volume is reduced due to the pinhole before the spherical mirror. R2 has a half-confocal resonator geometry, so the radius of the beam at the flat mirror is expected to be a factor of $\sim \frac{1}{\sqrt{2}}$ of the radius at the spherical mirror, which is $r \sim 150\,\text{µm}$ due to the pinhole. Using similar cones, the effective volume of the gain medium $V_{\text{eff}}$ that is pumped is given by,

$$V_{\text{eff}} = \frac{\pi}{6}\left(\frac{r}{L}\right)^2 \left(\sqrt{2} - 1\right)^2 \left\{ \left(\frac{L}{\sqrt{2}-1} + l_2\right)^3 - \left(\frac{L}{\sqrt{2}-1} + l_1\right)^3 \right\} \qquad (5)$$

where $l_1$ is the separation of the flat mirror and the closest edge of the pumped volume, and $l_2$ is the separation of the flat mirror and the furthest edge of the pumped volume. For $l_1 \sim 4.5\,\text{mm}$, $l_2 \sim 12.5\,\text{mm}$ and $L \sim 26\,\text{mm}$, $V_{\text{eff}}(\text{R2}) \sim 0.365\,\text{µL}$.

The absorption of the pump by the gain medium depends on the optical path length of the gain medium in the direction of the pump. For R2, the mean optical path length $l_{\text{mean}}$ can be estimated by calculating the mean path length across a cylinder with the same length and volume as the effective volume of the gain medium, so

$$l_{\text{mean}} = \frac{4}{\pi} \sqrt{\frac{V_{\text{eff}}}{\pi(l_2 - l_1)}} \qquad (6)$$

For R2, $l_{\text{mean}} \sim 153\,\text{µm}$. For R1, $l_{\text{mean}} = 2\,\text{mm}$, which is the depth of the cuvette chamber.

We collected the light transmitted through the output coupler with either an unoccluded lens (Newport, KPX112) or a lens coupled with an iris diaphragm (Thorlabs, LA1484-A, and Thorlabs, SM1D12CZ) such that the f/numbers were 300 mm/22.9 mm = 13.1 for R1 and 300 mm/5.5 mm = 54.5 for R2, respectively. A custom-built spectrometer combining a photomultiplier tube (PMT) (Hamamatsu Photonics, R1166) connected to a high voltage supply (Hamamatsu Photonics, C9525) and a Czerny-Turner monochromator (Bentham, M300EA) with a focal length of 300 mm, f/number of 4.2 and a 1200 grooves $\text{mm}^{-1}$ grating, recorded the output intensity. Before the spectrometer, the intensity could be optically attenuated using UV fused-silica reflective neutral density filters with ODs ranging from 0.5 to 3.0 for R1 and 0.5 to 4.0 for R2 (Thorlabs) mounted in a filter wheel working in conjunction with a OD 4.0 filter in a flip mount. Optical feedback was prevented by aligning the flip mount for R1 and R2 and the filter wheel for R2 so that the normal to each filter was not parallel to the direction of the incident light. For R1 the mounts were manual but for R2 they were replaced with motorised equivalents (Thorlabs, FW102C, and Thorlabs, MFF102/M). For R2, optomechanics prevented stray light from entering the spectrometer, and a dielectric longpass filter with a 500 nm cut-on (Thorlabs, FEL0500) connected to the light-collecting lens blocked any scattered pump light. For R2, the bias voltage to the PMT was –860 V because this enabled single-photon measurements without compromising linearity.

The voltage across a 50 Ω terminator (for R2, Pico Technology, TA051) connected to the PMT was recorded using an oscilloscope (for R1, Tektronix, TDS210 and for R2, Tektronix, TDS3052). The waveforms were transferred to a PC for real-time analysis in a custom program that controlled the optical assembly (National Instruments, LabVIEW 8).

The term "pump" refers to the radiant energy input into the resonant cavities in units of photons pulse$^{-1}$, which can be converted to a spectral irradiance of the resonant cavities via multiplication by $\hbar\omega$ $(16\,\text{mm}^2)^{-1}$ $(\delta\lambda(0.12\,\text{nm}))^{-1}$ $(4\,\text{ns pulse})^{-1}$, since the area of the cuvette window was 16 mm$^2$, the spectral

linewidth of the optical parametric oscillator was 0.12 nm and the optical pulse length was 3 ns to 5 ns, and $\hbar\omega$ is the energy of one photon with a wavelength of 493 nm. The term "intensity" refers to the radiant energy emitted from the resonant cavities in the direction of the detection optics per pulse. The reported intensity is different to the signal observed by the detector by a factor that accounts for the optical attenuation of the spectrometer, and does not account for the radiation that emits in directions that bypass the spectrometer. The oscillation build-up in the resonant cavity would have been greater than the measured intensity. The output intensity from R2 was calibrated by measuring the response of the detection electronics to single photons, so it is reported in units of photons pulse$^{-1}$. Equivalently, the output intensity from R2 can be converted to a spectral intensity emitted from the resonant cavity with units $\hbar\omega$ $(1.3 \times 10^{-3}\,\text{sr})^{-1}$ $(\delta\lambda(0.4\,\text{nm}))^{-1}$ pulse$^{-1}$, where the solid angle of the spectrometer is $1.3 \times 10^{-3}\,\text{sr}$ and the linewidth of the monochromator is approximately 0.4 nm, and $\hbar\omega$ is the energy of one photon at the selected wavelength of the monochromator. For R2, optomechanics and a filter prevented light from external sources to the resonant cavity from entering the spectrometer (more details in "Free-space optical assembly"), but this was not implemented for R1. Consequently, R1 could not be calibrated by measuring the response of the detection electronics to single photons. Instead, the intensity has been reported in arbitrary units proportional to the number of photons emitted from the resonant cavity in the direction of the detection optics per pulse.

**Description of further experiments in R2.** Two additional experiments were performed in R2 using the same dye-labelled M13 sample to test whether photobleaching could cause the step-changes in the intensity observed in Fig. 4 (Fig. 5, Supplementary Fig. 7 and 10). Further experiment 1 was similar to the experiment described in Results, except that 91 fmol mL$^{-1}$ mouse IgG2a isotype control, which should not be bound by M13, was added to the gain medium before 91 fmol mL$^{-1}$ cp-mAb was added, which was then increased to 1.9 pmol mL$^{-1}$. Buffer was not added to the reservoir between measurements and the antibody was allowed to mix for at least 12 min prior to measurement. Further experiment 2 was similar to the previous experiments, except that the dye-labelled M13 was diluted by a factor of ~0.89 in type 1 water and the final sample volume was ~2.75 mL, which was less than the 6.85 mL used in previous experiments, and the in-line filter was removed. Also, prolonged storage of the dye-labelled M13 at 4 °C resulted in some sedimentation, so the solution was transferred to a fresh container and then centrifuged to remove any residual insoluble material. For this experiment, only 4.5 pmol mL$^{-1}$ cp-mAb was added. The changes in the threshold dynamics with time before and after the addition of antibody were measured in both experiments. Emission spectra before and after the addition of antibody were acquired for further experiment 2 but not for further experiment 1.

**Characterisation of laser properties.** The threshold dynamics were recorded by measuring the output intensity at a fixed wavelength at variable pump levels. The intensity of the pump pulses were controlled by rotating the first polarizer. If the intensity was outside of the linear range of the spectrometer, the optical attenuation was adjusted to compensate and the measurement was repeated. In R1, the wavelength was fixed to 527.0 nm and 528.0 nm for the measurements of fluorescein and dye-labelled M13, respectively. In R2, the wavelength was fixed to 520.5 nm and 523.5 nm for the measurements of fluorescein and dye-labelled M13, respectively, except for further experiment 2, for which the wavelength was fixed to 525.0 nm.

The spectral measurements were acquired by fixing the pump level and recording the intensity at different emission wavelengths by rotating the monochromator grating.

**Characterisation of optical properties.** The absorption spectra were recorded using a UV-Vis spectrophotometer (for experiments using R1, Hitachi, Digilab U-1800; for experiments using R2, Thermo Scientific, Nanodrop 2000C) with 1 cm path length cuvettes.

The mass of a single M13 is taken to be 16.4 MDa ± 0.6 MDa (error is 2 standard deviations from the mean) which is the reported mass of fd virus based on measurements of the translational diffusion coefficient, sedimentation coefficient and density increment.[28] The extinction coefficient of M13 is taken to be $3.84 \pm 0.06\,\text{mg}^{-1}\,\text{cm}^2$ at 269 nm (error is 95% confidence limit) which is the reported extinction coefficient of fd virus in KCl/P buffer.[54] Based on these values, the extinction coefficient used to determine the concentration of M13 was $6.30 \times 10^7\,\text{M}^{-1}\,\text{cm}^{-1}$ at 269 nm.

The absorption of both the M13 and the fluorescein at both the dye peak and at 269 nm were accounted for in the calculations of the concentration of dye-labelled M13 and the number of attached dyes (Supplementary Fig. 2). The precision of the stated concentrations was limited by the precision of the pipettes used (Gilson, Pipetman) and the precision of the UV-Vis spectrophotometers.

For samples measured in R2, the extinction coefficient of 34.0 µg mL$^{-1}$ fluorescein, 100 mM sodium borate, pH 9.0 was calculated to be 82752 M$^{-1}$ cm$^{-1}$ at the peak absorption at 490 nm and 977 M$^{-1}$ cm$^{-1}$ at 269 nm; the extinction coefficient of M13 at the absorption peak for the attached dyes at 494 nm was

$1.3 \times 10^6\,\mathrm{M^{-1}\,cm^{-1}}$ (Supplementary Fig. 2). The extinction coefficients of fluorescein attached to M13 were assumed to be the same as for unattached fluorescein at 269 nm and at the absorption peak. The concentration of the dyes attached to the dye-labelled M13 was $12.8\,\mathrm{nmol\,mL^{-1}}$ and the concentration of dye-labelled M13 was $23\,\mathrm{pmol\,mL^{-1}}$, so there were 564 dyes per M13 on average. The fluorescein was diluted by a factor of 0.13 before further measurements were acquired, so its final concentration was $13.3\,\mathrm{nmol\,mL^{-1}}$.

For samples measured in R1, the extinction coefficient of $100\,\mu\mathrm{g\,mL^{-1}}$ fluorescein, 100 mM sodium borate, pH 9.0 was calculated to be $82413\,\mathrm{M^{-1}\,cm^{-1}}$ at the peak absorption at 491 nm and $10634\,\mathrm{M^{-1}\,cm^{-1}}$ at 269 nm; the extinction coefficient of M13 at the absorption peak for the attached dyes at 493 nm was determined to be $1.4 \times 10^6\,\mathrm{M^{-1}\,cm^{-1}}$ (Supplementary Fig. 2). The concentration of the dyes attached to the dye-labelled M13 was calculated to be $159\,\mathrm{nmol\,mL^{-1}}$ and the initial concentration of dye-labelled M13 was $413\,\mathrm{pmol\,mL^{-1}}$, so there were 386 dyes per M13 on average. The initial concentration of fluorescein was $301\,\mathrm{nmol\,mL^{-1}}$.

For both R1 and R2, the wavelength of the pump was 493 nm. For R1, the extinction coefficients at 493 nm compared to the extinction coefficient at the absorption peaks were a factor of 0.976 and 1 for fluorescein and dye-labelled M13, respectively. For R2 these factors were 0.984 and 0.991, respectively.

The separation of fluorescein dyes in solution was calculated by taking the cube root of the mean volume per dye.

**Theoretical models.** The changes in the energy build-up in a resonator mode and in the upper state population of the dyes can be modelled using the two rate equations:

$$\frac{dn}{dt} = K_{\mathrm{RAD}} D_2 (n + n_0) + K_{\mathrm{RAD}} D_2 - K_L n \tag{7}$$

$$\frac{dD_2}{dt} = K_F \varphi (D - D_2) - K_{\mathrm{RAD}} D_2 (n + n_0) - K_2 D_2 \tag{8}$$

where $n$ is the energy in the resonator mode, $\varphi$ is the rate of pumping of the resonator mode, $D$ is the number of dyes in the resonator mode volume, $D_1$ and $D_2$ are the number of dyes in the lower and upper state of the laser transition, respectively, $K_F$ is the strength of the coupling of the pump with the dyes, $K_{\mathrm{RAD}}$ is the rate of emission into the mode per dye, $K_2$ is the rate of decay from the upper state per dye, $K_L$ is the rate of loss from the resonator and $n_0$ is the seed fluorescence.

In equation (7), the first and second term describe the rate of stimulated and spontaneous emission, and the third term describes the rate of loss from the resonator. In equation (8), the first term describes the rate of pumping of the dyes, and the second and third terms describe the depopulation of the upper state due to stimulated emission and all other sources, respectively.

The rate of pumping is derived from the Beer–Lambert law assuming that every pump photon absorbed results in a dye in the upper state:

$$\frac{dD_2}{dt} = \varphi \left(1 - e^{-\frac{D_1}{\nu} \varepsilon l}\right) \sim \varphi\, D_1 \frac{\varepsilon l}{\nu} = K_F \varphi (D - D_2) \tag{9}$$

where $\varepsilon$ is the extinction coefficient multiplied by ln(10), $l$ is the optical depth of the resonator mode and $\nu$ is the volume of the resonator mode that is being pumped. This approximation is valid for R2 because the concentration of the dyes was low and the path length across the resonant cavity mode was narrow. For R1, these assumptions do not strictly hold, but are necessary to simplify the equations so that they can be solved analytically. Additionally, the Beer–Lambert law assumes that the dyes are homogeneously distributed in the solution, which was not the case for gain media composed of dye-labelled M13.

A full description of the threshold dynamics would require a rate equation for the energy build-up in each resonator mode, but consideration of just a single mode is sufficient to model the key aspects of the system.[55] Consistent with this approach, the volume of the resonator mode being pumped $\nu$ is set equal to the effective volume of the gain medium being pumped $V_{\mathrm{eff}}$.

The rate of radiative emission per dye into the resonator mode is a function of the rate of spontaneous emission and the number of resonator modes:

$$K_{\mathrm{RAD}} = \frac{\gamma_{\mathrm{RAD}}}{\rho} \tag{10}$$

where $\gamma_{\mathrm{RAD}}$ is the rate of spontaneous emission per dye and $\rho$ is the number of resonator modes.

The rate of decay from the upper state per dye is the sum of the rate of spontaneous emission and the rate of non-radiative decay:

$$K_2 = \gamma_{\mathrm{RAD}} + \gamma_{\mathrm{NON}} \tag{11}$$

where $\gamma_{\mathrm{NON}}$ is the rate of non-radiative decay per dye.

$K_L$ is determined by the loss rate of the resonator, which is given by

$$K_L = -\frac{1}{\tau_r} \ln(1 - LOSS) \tag{12}$$

where $\tau_r$ is the round-trip time and $LOSS$ is the fraction of the oscillation build-up lost per round-trip.

The energy in the resonator mode is derived from equation (7) and equation (8) by assuming steady-state emission, eliminating $D_2$ and solving the resulting quadratic equation:

$$n = \frac{\beta}{2 K_{\mathrm{RAD}}} \left\{ K_F\, \varphi \left(\frac{D K_{\mathrm{RAD}}}{K_L} - 1\right) - K_{\mathrm{RAD}} n_0 - K_2 \right.$$
$$\left. + \sqrt{\left(K_F\, \varphi \left(\frac{D K_{\mathrm{RAD}}}{K_L} - 1\right) - K_{\mathrm{RAD}}\, n_0 - K_2\right)^2 + \frac{4 K_F \varphi\, D\, K_{\mathrm{RAD}}^2}{K_L}(n_0 + 1)} \right\} \tag{13}$$

where $\beta$ is the optical attenuation due to the output coupler and the spectral bandwidth of the spectrometer.

The build-up of energy in the dominant resonator mode depends on optical amplification, which implies the presence of a non-zero seed fluorescence $n_0$ prior to amplification. This is modelled as the energy in the resonator mode when there is no stimulated emission. From equation (7) and equation (8):

$$\frac{dn_0}{dt} = \theta\, \gamma_{\mathrm{RAD}}\, D_2 - K_L\, n_0 \tag{14}$$

$$\frac{dD_2}{dt} = K_F\, \varphi\, (D - D_2) - K_2\, D_2 \tag{15}$$

where $\theta$ is the fraction of the spontaneous emission that emits at a wavelength and in a direction such that it can contribute to the seed fluorescence. Using the same assumptions as for equation (13):

$$n_0 = \frac{D\, \theta\, \gamma_{\mathrm{RAD}}}{K_L \left(1 + \frac{K_2}{K_F\, \varphi}\right)} \tag{16}$$

Light may have entered the spectrometer from external sources to the resonant cavity, which is accounted for by a constant offset.

The threshold point $\varphi_{\mathrm{th}}$ is given by

$$K_F\, \varphi_{\mathrm{th}} \left(\frac{D K_{\mathrm{RAD}}}{K_L} - 1\right) - K_{\mathrm{RAD}} n_0 - K_2 = 0 \tag{17}$$

from equation (13). Including the term for the seed fluorescence, this is then

$$K_F\, \varphi_{\mathrm{th}} \left(\frac{D K_{\mathrm{RAD}}}{K_L} - 1\right) - K_{\mathrm{RAD}} \frac{D\, \theta\, \gamma_{\mathrm{RAD}}}{K_L \left(1 + \frac{K_2}{K_F\, \varphi_{\mathrm{th}}}\right)} - K_2 = 0 \tag{18}$$

which can be solved to find

$$\varphi_{\mathrm{th}} = \frac{K_2 (2K_L - D K_{\mathrm{RAD}}) + D K_{\mathrm{RAD}} \left\{ \theta\, \gamma_{\mathrm{RAD}} + \sqrt{(\theta\, \gamma_{\mathrm{RAD}} - K_2)^2 + \frac{4\, \theta\, \gamma_{\mathrm{RAD}}\, K_2\, K_L}{D K_{\mathrm{RAD}}}} \right\}}{2 K_F (D K_{\mathrm{RAD}} - K_L)} \tag{19}$$

From equation (11), this can be simplified by noting that

$$\theta\, \gamma_{\mathrm{RAD}} \ll K_2 = \gamma_{\mathrm{RAD}} + \gamma_{\mathrm{NON}} \tag{20}$$

Since

$$\theta \ll 1 \tag{21}$$

and that for gain to exceed losses

$$\frac{D K_{\mathrm{RAD}}}{K_L} > 1 \tag{22}$$

so that the threshold point is given by

$$\varphi_{\mathrm{th}} = \frac{K_2 K_L}{K_F (D K_{\mathrm{RAD}} - K_L)} \tag{1}$$

The minimum number of dyes required to achieve threshold $D_{\mathrm{min}}$ even at very large levels of pumping, is given by,

$$D_{\mathrm{MIN}} = \frac{K_L}{K_{\mathrm{RAD}}} \tag{23}$$

From equation (13), the output energy far above threshold when $\varphi \to \infty$ is given by

$$\frac{n}{\varphi} \to \frac{\beta K_F}{K_{\mathrm{RAD}}} \left(\frac{D K_{\mathrm{RAD}}}{K_L} - 1\right) \tag{2}$$

Furthermore, if $D \gg D_{\mathrm{min}}$, then

$$\frac{n}{\varphi} \to \frac{\beta K_F D}{K_L} \tag{24}$$

and the gain above threshold ceases to depend on the rate of spontaneous emission.

The time-dependence of the pump energy was accounted for by assuming a Gaussian profile with a FWHM of 4 ns,

$$\varphi(t) = \varphi_0 \times \mathrm{Gaussian}(t) \tag{25}$$

and calculating the output energy based on the peak pump energy. The non-linear response of output energy to increased pumping means that the peak pump energy has the greatest contribution to the overall output energy. An alternative method was tested in which the pump pulse was split into sub-nanosecond time bins over a ~50 ns time window centred about the pulse peak and the output energy calculated by summing the output energies from each time bin, but the results generated were negligibly different to the preferred method. The calculation of the threshold point accounted for the time-dependence of the pump energy.

For R2, only $\frac{1}{2\,\mathrm{mm}}$ of the pump impinged on the resonator mode because its diameter was less than the 2 mm height of the window. The remainder would have pumped dyes outside of the dominant resonator mode.

The threshold point for each threshold measurement could be readily determined using a model similar to equation (13),

$$n = \chi_0 + \chi_1 \left\{ \left( \frac{\varphi}{\varphi_{\mathrm{th}}} - 1 \right) + \sqrt{ \left( \frac{\varphi}{\varphi_{\mathrm{th}}} - 1 \right)^2 + \chi_2 \varphi } \right\} \tag{26}$$

where $\chi_n$ are fit parameters that are not directly linked to the experimentally observable variables, $\varphi_{\mathrm{th}}$ is the threshold point and the time-dependence of the pump is disregarded.

**Data analysis**. For R1, the threshold dynamics of fluorescein and dye-labelled M13 were analysed by globally fitting each set of data with a model that employed equation (13) as well as equation (16) for the seed fluorescence (Fig. 2a, Supplementary Fig. 5). The parameters $\gamma_{\mathrm{RAD}}$, $K_L$, $K_F$, $l_{\mathrm{mean}}$, and $V_{\mathrm{eff}}$ were all fixed and the same for each set of measurements. Each of these were calculated in advance (see Methods sections "Free-space optical assembly" and "Theoretical models"), except for $\gamma_{\mathrm{RAD}}$ which is known to equal 0.240 ns$^{-1}$.[56] $D$ was fixed but different for each set of measurements. A factor was calculated from the spectral measurements to account for the difference between the intensity at the measured emission wavelength and the spectral peak, which was fixed for each set of measurements except for 146 nmol mL$^{-1}$ fluorescein, since there was no spectral measurement of this sample and it may have undergone a spectral shift. The parameters $\rho$, $\beta$, $\theta$ and a constant offset to account for external light sources were not fixed and were shared between each set of data because they should be consistent between measurements. The parameter $\gamma_{\mathrm{NON}}$ was not fixed and shared between the sets of dye-labelled M13 measurements, but allowed to vary individually for each set of fluorescein measurements.

In Fig. 3a, the line representing the threshold point as a function of probe concentration was simulated using this model. In Fig. 3b, the pump energy values were selected because they represent local minima in a plot of pulse energy variability against pulse energy. The pulse energy varies from pulse to pulse due to variability in the intensity of the emission from the optical parametric oscillator. The lines were simulated using the same model.

For R1, the threshold measurements were additionally fit using equation (26) to accurately determine the position of the threshold point (Fig. 3a, Supplementary Fig. 6). From equation (1), as $D \rightarrow D_{\mathrm{MIN}}$, $\varphi_{\mathrm{th}} \rightarrow \infty$ and if $D < D_{\mathrm{MIN}}$ then $\varphi_{\mathrm{th}} < 0$, so the parameter $\varphi_{\mathrm{th}}$ was substituted for its reciprocal in the fitting function and subsequently derived after the model converged. $\varphi_{\mathrm{th}} < 0$ for the most dilute dye-labelled M13 sample, implying that the probe concentration was insufficient to achieve lasing.

For R2, the threshold measurements were individually fit using equation (26) (Fig. 4b, Supplementary Fig. 8). For the further experiments in R2, the threshold measurements were also fit using equation (26), but parameters $\chi_1$ and $\chi_2$ were shared between sets of data from the same experiment (Fig. 5, Supplementary Fig. 10). From comparison of the parameters in equation (26) and equation (13) with zero seed fluorescence, we expected $\chi_1$ and $\chi_2$ to be consistent between these measurements.

For R1, when the pumping was above the threshold for lasing, the spectra were fit to a model consisting of a sum of two Gaussian functions with a constant offset (Fig. 2b and inset, Supplementary Fig. 4). For R2, the above-threshold spectra were fit to a model consisting of a single Voigt function or a sum of two Voigt functions with no constant offset, and the below-threshold spectra were fit to a Lorentzian function with no constant offset. In Fig. 4, the scatter points in the spectra are the means of four measurements at each wavelength; see Supplementary Fig. 7 for the same data presented without averaging. The mean and coefficient of variation of the pump energy has been reported for each emission spectrum (Supplementary Fig. 4, Supplementary Fig. 7). The spectral peak centres and linewidths were determined by the model fitting and the reported errors are standard errors.

**Reporting summary**. Further information on research design is available in the Nature Research Reporting Summary linked to this article.

## Data availability
Source data are provided as a Source Data file. The source data are also available for download from UCL's open-access repository, UCL Discovery [http://discovery.ucl.ac.uk/].

## Code availability
The custom computer code for the fitting functions used for the data analysis is provided as a Computer Code file. The computer code is also available for download from UCL's open-access repository, UCL Discovery [http://discovery.ucl.ac.uk/].

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

## Acknowledgements

We thank S. Chu for discussions during and after J.E. Hales' PhD viva and C.W.M. Kay for use of equipment and laboratory space. We acknowledge the London Centre for Nanotechnology, which is the institution where the free-space optical assembly was constructed and where the characterisation of the laser properties was undertaken. This work was supported by the Wellcome Trust (PhD studentship to J.E.H.), the Bioprocessing Research Industry Club (BBSRC, BB/E005942/1), the EPSRC (EP/H026622/1 and EP/M009564/1), the BBSRC Excellence with Impact funding to UCL, and UCL Business (Proof-of-concept funding).

## Author contributions

J.E.H. constructed the virus laser with supervision from G.M., prepared samples with supervision from J.M.W. and P.A.D., planned experiments with G.A. and G.M. with input from J.M.W. for R1, and from P.A.D. for R2, carried out the experiments, constructed the theoretical models, analysed and visualised the data with guidance from G.A. and wrote the manuscript with G.A., which was read and commented on by G.M., J.M.W. and P.A.D.

## Additional information

**Competing interests:** Research was partially funded by UCL Business, who have a financial interest in the commercial success of virus lasers. J.E.H., J.M.W. and G.A. are co-authors on a patent application (WO 2013/093499) and J.E.H., G.M., P.A.D. and G.A. are co-authors on a second patent application whose respective values could increase if the methods and ideas described in this paper find widespread application.

