## [Peer Review File · Nature Communications]

Reviewers' comments:

Reviewer #1 (Remarks to the Author):

As I have pointed out in the first round of review, the method described in this manuscript is novel, and the data is solid. Most of the comments pointed out previously have been handled. I therefore would like to suggest the publication of this paper in Nature Communications.

But since it is only a proof-of-concept demonstration, in which only purified monoclonal antibody in simple solvents is tried, there is no evidence to show if it will work in complex environment (such as cellular milieu or blood serum) because of non-specific binding and other sources of noise. I therefore still believe it is an over claim to have this kind of conclusion:

“Viral-lasing detection probes are the first probes which can bind the full range of biomolecules targeted in clinical assays, including proteins, nucleic acids and cells, whilst also generating a lasing signal in an optical configuration compatible with ordinary fluorimeters.” (Page 4, line 10).

The authors need to avoid this kind of strong conclusion before they can demonstrate it. The complexity of real biological or clinical samples cannot be sufficiently replicated by just one specific control experiment (the mouse IgG2a isotype control in a simple solvent), and lasing threshold event is sensitive to environment changes that can introduce gain or loss.

Reviewer #2 (Remarks to the Author):

I'm happy with the changes the authors have made to my comments on the prior submission and am happy to recommend publication. As I previously indicated this is certainly more suited for Nature Comms, than Nature Nanotech. In terms of the open data, I think the data can be associated with the paper on publication, so while the UCL option is fine, I see no reason why this cannot be submitted with the paper to directly link with the published work.

Reviewer #4 (Remarks to the Author):

I had the opportunity to read the previous referees' comments and authors' replies, along with the revised manuscript. I agree with the main concern raised by Referee 1 that the demonstrated protocol in its current form is unlikely to be effective for testing clinical samples that contain many biomolecules besides the target. The reason is not so much about non-specific binding but the expected contribution of the non-target molecules to the signal. Any optical absorption by non-target molecules will increase laser threshold. Therefore, this influence should be kept low and must be calibrated out in order to relate the measured threshold to the amount of target molecules. The authors seem to have ideas how to solve this issue. I think it is important that the authors test their idea and demonstrate that their approach indeed can work with a clinically relevant sample.

The main novelty of this manuscript lies in the use of M13 phages carrying both a specific target binding site and a large number of dye molecules per particle. The latter allowed for several hundred-fold improvement in the sensitivity to target molecules. This nice idea should be published somewhere.

On the other hand, the idea of using the sharp transition of intensity across laser threshold for sensing and imaging is not new. It is true that the signal response to probe is greatly enhanced near threshold. However, the sensitivity to any undesirable perturbation or error sources, such as absorption by non-target molecules, photobleaching of the probe, and uncalibrated changes in optical cavity quality factor, are also high, as Referee 3 pointed out. In this sense, the claim of 5-decade increase in signal-to-noise ratio (SNR) is misleading (unless the noise is generated downstream of the bio-laser; for example, photodetector noise). One may argue that there is no real SNR advantage.

The authors seem to be optimistic about improving the detection sensitivity further (Lines 12-16, Page 9). It is however not very convincing. Any further improvement in sensitivity and accuracy will increase the impact of this manuscript.

The effect of probe-target binding on threshold (e.g. Fig. 4c) is interesting, although the quality of the data is marginally good. The authors explain the mechanism in terms of a redshift of dyes and energy transfer. I find this explanation not quite convincing. It will be helpful to provide any experimental support, such as direct observation of red shift, and calculation to show that the change can account for the measured threshold change.

Viral lasers for biological detection: Authors' response to referees' comments.

Reviewer #1 (Remarks to the Author):

As I have pointed out in the first round of review, the method described in this manuscript is novel, and the data is solid. Most of the comments pointed out previously have been handled. I therefore would like to suggest the publication of this paper in Nature Communications.

But since it is only a proof-of-concept demonstration, in which only purified monoclonal antibody in simple solvents is tried, there is no evidence to show if it will work in complex environment (such as cellular milieu or blood serum) because of non-specific binding and other sources of noise. I therefore still believe it is an over claim to have this kind of conclusion:

“Viral-lasing detection probes are the first probes which can bind the full range of biomolecules targeted in clinical assays, including proteins, nucleic acids and cells, whilst also generating a lasing signal in an optical configuration compatible with ordinary fluorimeters.” (Page 4, line 10).

The authors need to avoid this kind of strong conclusion before they can demonstrate it. The complexity of real biological or clinical samples cannot be sufficiently replicated by just one specific control experiment (the mouse IgG2a isotype control in a simple solvent), and lasing threshold event is sensitive to environment changes that can introduce gain or loss.

We are very grateful to the reviewer for the suggestion to publish the paper in Nature Communications, as well as for pointing out the need to more precisely delineate between the potential of this technology and the progress we have made so far. Thus the sentence in question has been modified to read:

“Dye-labelled phage-clones derived from phage-display are already known to maintain their target-binding capability after selective labelling, and so these probes can replace dye-conjugated primary antibodies in biological assays (31). We have shown that probes of this type can be used as viral-lasing media to generate a lasing signal in an optical configuration compatible with ordinary fluorimeters, and also to detect the binding of a target antibody molecule.”

Reviewer #2 (Remarks to the Author):

I'm happy with the changes the authors have made to my comments on the prior submission and am happy to recommend publication. As I previously indicated this is certainly more suited for Nature Comms, than Nature Nanotech. In terms of the open data, I think the data can be associated with the paper on publication, so while the UCL option is fine, I see no reason why this cannot be submitted with the paper to directly link with the published work.

We thank the referee for the positive comments. In addition, we are happy to associate the data directly with the paper on publication in addition to depositing the data on UCL Discovery.

Reviewer #4 (Remarks to the Author):

I had the opportunity to read the previous referees' comments and authors' replies, along with the revised manuscript. I agree with the main concern raised by Referee 1 that the demonstrated protocol in its current form is unlikely to be effective for testing clinical samples that contain many biomolecules besides the target. The reason is not so much about non-specific binding but the expected contribution of the non-target molecules to the signal. Any optical absorption by non-target molecules will increase laser threshold. Therefore, this influence should be kept low and must be calibrated out in order to relate the measured threshold to the amount of target molecules. The authors seem to have ideas how to solve this issue. I think it is important that the authors test their idea and demonstrate that their approach indeed can work with a clinically relevant sample.

We are grateful to the reviewer for raising this issue, which affects any biochemical assay, and note that the usual mitigation strategies are available. For example, different phage-clones with the same density of fluorescein attachments but with binding or non-binding g3p coat proteins could be used to create a differential measurement. This will take advantage of both the programmability of phage and their binding specificity.

In addition, we have included a calculation and a short discussion on the impact of optical absorption by non-target molecules in which we propose simply diluting the sample to overcome this problem, as is commonly practiced for other assays (p19, lines 9-20).

The reviewer is correct to point out the importance of developing and demonstrating detailed procedures, based on such ideas, for clinical use, and this is indeed what we plan for the next phase of our research. The purpose of the current paper is to make the scientific community aware of a new opportunity for biomolecular recognition and to describe the underlying physical and biochemical principles, and we have modified the introductory section and conclusion to make this clear to avoid any confusion.

The main novelty of this manuscript lies in the use of M13 phages carrying both a specific target binding site and a large number of dye molecules per particle. The latter allowed for several hundred-fold improvement in the sensitivity to target molecules. This nice idea should be published somewhere.

On the other hand, the idea of using the sharp transition of intensity across laser threshold for sensing and imaging is not new. It is true that the signal response to probe is greatly enhanced near threshold. However, the sensitivity to any undesirable perturbation or error sources, such as absorption by non-target molecules, photobleaching of the probe, and uncalibrated changes in optical cavity quality factor, are also high, as Referee 3 pointed out. In this sense, the claim of 5-decade increase in signal-to-noise ratio (SNR) is misleading (unless the noise is generated downstream of the bio-laser; for example, photodetector noise). One may argue that there is no real SNR advantage.

We appreciate the referee's compliments concerning the idea in our paper, and in particular for recognizing that it provides for a several hundred-fold improvement in sensitivity in

fluorescence-only assays compared to equivalent assays employing dye-labelled antibodies. There is thus a clear SNR advantage for detecting small concentrations of target molecules. The five decades of improvement in SNR does refer, as suggested by the reviewer, to noise generated downstream of the bio-laser. We are grateful to the reviewer for highlighting that our statement could be misinterpreted and have modified it accordingly.

An output light intensity > 100,000-times greater than below threshold represents a real advantage for a measurement system. Analytical instruments for low concentration assays typically require highly sensitive detectors such as photomultiplier tubes for recording single photons, which introduces optical shot noise. In a viral laser assay, the photon flux is so much greater that the relative contribution of optical shot noise would be negligible, meaning that there is a real SNR advantage. Furthermore and very important for lowering the cost of practical measurement apparatus, a PIN photodiode could be used instead of an expensive single-photon detection system.

The reviewer correctly identifies the many factors which can introduce systematic errors. They can be largely eliminated via the differential measurement technique described above, which is in exact analogy to what is accepted practice for surface plasmon resonance, considered the “gold standard” for binding assays. The accuracy will be given by the precision with which lasing thresholds can be measured, which is very high given our ability to measure photon fluxes for the pump, and the ability to create matched solutions of binding and non-binding phages to which the biological fluid of interest would be added for differential analysis. We have included further analysis which suggests that it would be possible to deconvolve perturbations due to error sources from shifts caused by binding (p16, lines 21-25).

The authors seem to be optimistic about improving the detection sensitivity further (Lines 12-16, Page 9). It is however not very convincing. Any further improvement in sensitivity and accuracy will increase the impact of this manuscript.

We thank the reviewer for highlighting that further improvements in sensitivity and accuracy would increase the impact of the manuscript. We have now introduced a sentence to emphasise that the strategies noted for improving the detection sensitivity are in fact utilised in the design of resonant cavity R2, and that extending this approach can yield further performance enhancements (p13 line 23 to p14 line 1). For the experiments conducted in R2, the concentration of probes was 23 pmol/mL M13, which is lower than the minimum concentration required for achieving lasing in resonant cavity R1. The detection sensitivity (< 100 fmol/mL of protein) observed in the current experiment is already exceptional and implies a large potential impact for this manuscript.

The effect of probe-target binding on threshold (e.g. Fig. 4c) is interesting, although the quality of the data is marginally good. The authors explain the mechanism in terms of a redshift of dyes and energy transfer. I find this explanation not quite convincing. It will be helpful to provide any

experimental support, such as direct observation of red shift, and calculation to show that the change can account for the measured threshold change.

We are grateful to the referee for highlighting this point and have updated the section accordingly. A red shift on binding to fluorescein has been directly observed by other groups : anti-fluorescein fab fragments bound to fluorescein shift the absorption peak from 491 nm to the ranges 505 nm to 507 nm and/or 518 nm to 520 nm with fluorescence quenching maxima values of 95 % to 97 %. We also cite a comprehensive theoretical analysis by Aas *et al* which calculates the impact of acceptor dye concentration on threshold position in a FRET laser system.

We agree that further experiments are required to fully understand the mechanism and we plan to conduct such measurements in the future. However, this is the first observation of this effect and we think this will be of interest to both laser physicists and to the life sciences community.

In a typical fluorescent system the dyes emit independently of the other dyes, but in a laser system all of the dyes in the dominant mode volume contribute to the same laser signal. Therefore, the signal from a laser is intrinsically more sensitive to changes in just a few dyes since they all contribute to the measured signal. We have cited a paper by Ma *et al* in which adding just 1 acceptor dye per 100 donor dyes on a rod-like virus decreases the amplitude of what we interpret to be the radiative decay component from 73 % to 10 %, which would increase K_2 in equation (1).

REVIEWERS' COMMENTS:

Reviewer #1 (Remarks to the Author):

The authors have addressed my concerns in the revision. I suggest its publication in Nature Comm.

Reviewer #2 (Remarks to the Author):

I remain comfortable that the work presented is both sufficiently interesting and sufficiently convincing that it is suitable for publication in Nature Comms. My view is that this is true even given the fact that this is not yet a convincing biomedical assay, and that there remain open questions as to whether it will be in the future, but the concept is strong and the further refinements have strengthened the case. I look forward to seeing the follow on studies exploring the practical applications of the work.

Response to referees' comments

Reviewer #1 (Remarks to the Author):

The authors have addressed my concerns in the revision. I suggest its publication in Nature Comm.

We are grateful to the reviewer for their suggestion to publish the manuscript in Nature Communications.

Reviewer #2 (Remarks to the Author):

I remain comfortable that the work presented is both sufficiently interesting and sufficiently convincing that it is suitable for publication in Nature Comms. My view is that this is true even given the fact that this is not yet a convincing biomedical assay, and that there remain open questions as to whether it will be in the future, but the concept is strong and the further refinements have strengthened the case. I look forward to seeing the follow on studies exploring the practical applications of the work.

We thank the reviewer for this positive assessment of the manuscript.